# DG-GAN: A High Quality Defect Image Generation Method for Defect Detection

**DOI:** 10.3390/s23135922

**Published:** 2023-06-26

**Authors:** Xiangjie He, Zhongqiang Luo, Quanyang Li, Hongbo Chen, Feng Li

**Affiliations:** 1School of Automation and Information Engineering, Sichuan University of Science and Engineering, Yibin 644000, China; 321085404413@stu.suse.edu.cn (X.H.); 320085404308@stu.suse.edu.cn (Q.L.); 2Artificial Intelligence Key Laboratory of Sichuan Province, Sichuan University of Science and Engineering, Yibin 644000, China; 3Sichuan Shuneng Electric Power Company Ltd., Chengdu 610000, China; chb740921@163.com; 4School of Engineering and Technology, The Open University of Sichuan, Chengdu 610073, China; 5Engineering Research Center of Integration and Application of Digital Learning Technology, Ministry of Education, Beijing 100039, China

**Keywords:** deep learning, generating adversarial networks, defect image generation, defect detection

## Abstract

The surface defect detection of industrial products has become a crucial link in industrial manufacturing. It has a series of chain effects on the control of product quality, the safety of the subsequent use of products, the reputation of products, and production efficiency. However, in actual production, it is often difficult to collect defect image samples. Without a sufficient number of defect image samples, training defect detection models is difficult to achieve. In this paper, a defect image generation method DG-GAN is proposed for defect detection. Based on the idea of the progressive generative adversarial, D2 adversarial loss function, cyclic consistency loss function, a data augmentation module, and a self-attention mechanism are introduced to improve the training stability and generative ability of the network. The DG-GAN method can generate high-quality and high-diversity surface defect images. The surface defect image generated by the model can be used to train the defect detection model and improve the convergence stability and detection accuracy of the defect detection model. Validation was performed on two data sets. Compared to the previous methods, the FID score of the generated defect images was significantly reduced (mean reductions of 16.17 and 20.06, respectively). The YOLOX detection accuracy was significantly improved with the increase in generated defect images (the highest increases were 6.1% and 20.4%, respectively). Experimental results showed that the DG-GAN model is effective in surface defect detection tasks.

## 1. Introduction

During the process of industrial manufacturing, if the industrial product has not been detected for surface defects, the quality division of the product will be biased. If defective products enter the market, they will directly affect the reputation of the products and the economic benefits of the production enterprises. In addition, if some important components have not been inspected for surface defects to determine their quality level, the direct use of these components can lead to major safety incidents and may even lead to the shutdown of the entire industrial production. Therefore, surface defect detection is of great significance to the quality control of industrial products and the improvement of production efficiency. The surface defect detection of industrial products and various components is essential.

Traditional surface defect detection algorithms, such as LBP [1,2], FIR [3,4], and TEXTEM [5,6], first process the obtained defect images by means of image preprocessing (including image grayscale, image denoising, etc.). Then, machine learning algorithms such as Support Vector Machine (SVM) [7] or Decision Tree [8] are used to identify and classify processed defect images to finally achieve the purpose of defect detection. Although the traditional method has the advantages of low defect requirements, fast detection speed, and less time incurred, it has the disadvantages of low intelligence, narrow application scope, and a large limitation of judging defects.

In recent years, with the continuous improvement and deepening of deep learning theory, visual tasks based on deep learning have emerged endlessly, and surface defect detection is one of the hottest research focuses. Network models based on various deep learning methods, such as the Convolutional Neural Network (CNN) [9], Deep Belief Network (DBN) [10], Recurrent Neural Network (RNN) [11], Auto Encoder (AE) [12], and Generative Adversarial Network (GAN) [13], have been widely applied to various surface defect detection tasks. They have achieved excellent performance. The defect detection model, which is based on deep learning methods, can automatically extract features and achieve high-precision detection efficiently in real-time. Currently, deep learning defect detection methods require massive defect image samples as training data. However, in actual production, it is very difficult to manually collect enough defect image samples to be used as the training dataset for the detection model. If the training data is insufficient, there will be an overfitting phenomenon in the training of the model, thus resulting in low detection accuracy. As a result, it is difficult to collect defect image samples and insufficient data, which is one of the biggest problems faced by current deep learning defect detection methods.

To solve the problem of missing defect image samples in surface defect detection tasks, researchers first thought of applying Data Augmentation [14] to defect detection. First, the defect image dataset was expanded by image scaling, image rotation, image flipping, and image brightness adjustment. This method can improve the training effect of the model to a certain extent, but it also has certain defects. Because the data distribution of image samples enhanced by the data is not significantly different from that of the original image, the effect of network training is not considerably improved, and there is even the possibility of overfitting. In recent years, with the improvement and deepening of deep learning theory, scientists have proposed a variety of generative models that can be used to generate defect images, among which the most typical are the Variational Auto Encoder (VAE) [15] and Generative Adversarial Network (GAN) [16]. GAN networks and their variants are one of the most popular deep learning methods used in recent years. Although the GAN network has good generation performance, it still has some shortcomings, such as mode crashing and unstable training. Later, various GAN network variants were proposed, such as CGAN [17], DCGAN [18] and WGAN [19], to alleviate some of the shortcomings. But in the specific generation task, there are still many problems, such as the quality and diversity of the generated images, which are difficult to control. The advantages and disadvantages of some typical generation methods are listed in Table 1. Moreover, in the task of defect image generation, there are many kinds of defects; some defects themselves are not significantly different from their background, and some defects occupy a small proportion of the image. Extracting these features and balancing the quality and diversity of the generated images constitute a difficult problem faced by current generation methods.

In view of the above difficulties and problems, this paper proposes an improved generative adversarial network, DG-GAN to solve the problem of the lack of defect image samples in the task of surface defect detection by generating defect image samples. Through the study of existing defect image samples and defect-free image samples, the idea of progressive generation antagonism was used to improve the training stability of the network. In addition, D2 adversarial loss function, cyclic consistency loss function, a data augmentation module, and a self-attention mechanism were used to improve the network’s generation capabilities, thus ultimately producing defect images with high quality and high diversity. Among them, the data augmentation module makes the model training more stable, the self-attention mechanism makes the model have a global receptive field, and the D2 adversarial loss function and cyclic consistency loss function allow the model to learn more features from a small number of samples. This model does not require too much manual intervention and can independently learn complex sample features, balance the quality and diversity of generated images, and generate pseudo-defect images that are similar to real defect images, thereby effectively solving the problem of missing defect image samples. This paper contains the following three contributions:(1)In order to solve the problem of lack of defect image samples in surface defect detection task, a defect image generation network, DG-GAN, was proposed.(2)The proposed DG-GAN model can generate a high quality and high diversity of defect image samples, improve the detection accuracy of surface defect detection models, and facilitate the success of surface defect detection tasks.(3)For the DG-GAN model, a NEU surface defect dataset of a hot-rolled strip steel and a self-collected insulation partition defect dataset were used to improve the accuracy of the surface defect detection model.

The remainder of this article is organized as follows. Section 2 briefly summarizes the relevant research and achievements in the field of defect detection and Generative Adversarial Networks at home and abroad. Section 3 introduces the basic structure of the DG-GAN and its modules. In Section 4, comparison and ablation experiments of the proposed methods are carried out, and the experimental results are analyzed. In Section 5, a case study of defect detection on an extended dataset is presented. Finally, Section 6 summarizes all the work of this paper.

## 2. Related Research

### 2.1. Defect Detection

Since the initial development of defect detection in the 1980s, it has gone through nearly half a century of development, from traditional statistical methods [1,2], filtering method [3,4] and model methods [5,6] to deep learning methods after 2012. During this period, defect detection methods have emerged endlessly and achieved excellent results. These approaches have laid a solid foundation for applying defect detection in practice. It is mentioned in the literature [20] that defects in computer vision tasks tend to be an empirical concept of human beings rather than a purely mathematical definition. At present, there are mainly two methods of defect detection. One is based on supervised learning [21]; the other is based on unsupervised learning [22]. In the defect detection method based on the supervised learning mode, the model is trained by using manually marked defect image samples, in which the labels include the defect category, position coordinate, and rectangular box. However, the defect detection method based on unsupervised learning only needs to train defect image samples without labels to achieve the purpose of defect detection. This method can greatly reduce labor costs. The literature [23,24,25] has made a comprehensive and in-depth summary and elaboration of the defect detection methods based on deep learning and its optimization methods. These three papers reviewed in detail the theoretical development and evolution of defect detection technology based on deep learning and its optimization methods, as well as its practical research and application.

Defect detection technologies based on deep learning methods have been developed to date, among which the most mainstream are: the R-CNN [26] series, YOLO [27] series and SSD [28] series. In 2014, Ross et al. proposed the R-CNN target detection model. First, 1k–2k candidate regions were selected on the image, and then the candidate regions were sent to the feature extraction network to obtain the corresponding feature vectors. Then, the obtained feature vectors were sent to the SVM classifier for classification, and the position of the candidate boxes was modified. The method broke through the bottleneck in the field of target detection at that time, and the detection accuracy was at least 30% higher than before. It was then widely used in defect detection. In order to solve the problems of slow training speed and considerable space occupied by the R-CNN, Ross et al. [29] proposed the Fast R-CNN in 2015, which also uses the VGG16 [30] as the network backbone. Its training time is only one-ninth that of the R-CNN, and its test reasoning time is more than 200 times faster. The accuracy on the PASCAL VOC dataset increased from 62% with the R-CNN to 66% with the Fast R-CNN. The Faster R-CNN [31] is another masterpiece by Ross et al. As with the Fast R-CNN, the network also uses VGG-16 as the backbone, and the inference speed reached 5 fps on the GPU (including the generation of candidate regions). It can detect five images per second. The accuracy of the network was also further improved, and it won first place in several projects in the ILSVRC and COCO competitions in 2015.

Unlike the R-CNN series, the YOLO series is a one-stage regression method based on deep learning, while the R-CNN series is a two-stage classification method based on deep learning. Since its introduction in 2016, YOLO has been updated from the v1 version to the current v8 version, with each generation having optimizations and performance improvements over the previous generation. For example, YOLOV5 [32], based on YOLOV4 [33], introduced mosaic data enhancement, adaptive anchor frame calculation, and adaptive image scaling on the input side. The focus structure and CSP structure have been integrated into the benchmark network, and the FPN+PAN structure has been added to the neck network. In the head output layer, the training loss function GIOU_Loss and the DIOU_nms prediction box filtering have been improved. These improvements have greatly improved its speed and accuracy. In 2016, Liu et al. [28] proposed the SSD network, which uses a single signal to detect objects in images. For networks with input sizes of 300 × 300, 74.3% mAP and 59FPS scores were achieved with the VOC 2007 test set. For 512 × 512 networks, it achieved 76.9% mAP, thus surpassing the Faster R-CNN (73.2% mAP), which was the strongest at the time. In this paper, an advanced defect detection model was used to detect different surface defects in images accurately in real time. Table 2 shows the improvements and deficiencies of some classical defect detection models.

### 2.2. Generate Adversarial Networks

Goodfellow et al. [16] first proposed the Generative Adversarial Network in 2014. Since the GAN network was proposed, it has been widely used in medicine, computer vision, and speech processing. The literature [20] provides a comprehensive and in-depth summary of the research and application of the GAN network for defect detection. In addition, the literature [21] reviews the theoretical development and evolution of GAN networks in detail, as well as summarizes and classifies GAN-based defect detection technologies. The GAN network is based on the assumption of a zero-sum game between two people. The discriminator and generator train at the same time through the process of continuous confrontation, and finally generate a satisfactory result. Finally, the discriminator outputs a probability value of about 0.5. Although the GAN network has achieved remarkable success, it also has problems with unstable training and easy mode crashing. To solve these problems, researchers have proposed many variant networks, which, to some extent, have alleviated these problems, and the network performance has been further improved.

In 2018, Zhao et al. [34] proposed for the first time to apply the GAN network to perform defect detection. The idea was to learn defect features for repairing the input defect image via the GAN network, and then compare the defect sample with the repaired sample to locate the defect area in the image, so as to achieve the purpose of defect detection. The method achieved an average detection accuracy of 98.5323% in texture defect detection and 94.4253% in fabric defect detection. This method also took 1/3 less time than the previous method. The application of the GAN network in defect detection has implements the following two ideas. The first is to use the powerful generation capabilities of GANs to generate defect image samples, augment defect data sets, and improve the performance of defect detection models. Secondly, according to the generative antagonism of the GAN, the generator and discriminator are replaced by the defect detection network and the full convolutional discriminant network, respectively, so as to achieve the purpose of defect detection.

In 2019, Zhang et al. [35] proposed a Defect Generation Network, Defect-GAN, which generates realistic defects in image backgrounds with unique textures and appearances. The network uses a hierarchy-based structure to generate realistic defects and simulate the random variations of defects. In addition, it can flexibly control the position and category of defects generated within the image background. In the same year, Zhang et al. [36] proposed a Semi-Supervised Generative Adversarial Network (SSGAN) with two subnetworks to automatically detect defects in images, which were used to obtain more accurate pixel segmentation results. One was a segmentation network that divided defects from labeled and unlabeled images, which were based on a dual attention mechanism. The other was the Full Convolution Discriminant Network (FCD), which used two loss functions (adversarial loss and cross entropy loss) to generate a confidential density map of unlabeled images in semi-supervised learning. The method achieved an average intersection over union (IoU) of 79.0% and 81.8% in defect segmentation experiments using 1/8- and 1/4-labeled datasets, respectively. Moreover, the SSGAN is robust and flexible in segmentation under various scenarios.

Aiming at addressing the difficulty in collecting enough small and weak defects in practice, Niu et al. [37] proposed a defect image generation method with a controllable defect area and intensity. In this method, the defect area was treated as image repair using a generative adversarial network, and the defect mask was used to control the defect area. Based on the feature continuity between the defect and non-defect, the defect direction vector was constructed in the latent variable space, and the defect intensity was controlled to achieve a one-to-many correspondence between the defect mask and the image. This method greatly increases the quality and diversity of small and weak defect images.

GANs have great advantages in image generation and have achieved excellent performance; however, when there is a lack of data samples, and the foreground of the generated image is a small target (such as scratches and pit-spot defects proposed in this paper), the performance of the previous models have been poor, and the quality and diversity of generated defect images end up being poor. Therefore, the DG-GAN model proposed in this paper introduces a data augmentation module, a self-attention mechanism, a D2 adversarial loss function, and a cyclic consistency loss function to improve the generation capability of the network, generate high-quality and highly diverse defect images, and facilitate the success of defect detection tasks.

## 3. Methods

### 3.1. The Main Structure of the Model

In this paper, a defect image generation network model named DG-GAN was proposed, which contains two generators *G* and *C* with the same structure, and four discriminators $D1_{f}$, $D1_{i}$, $D2_{f}$, and $D2_{i}$ with the same structure. The DG-GAN network structure is illustrated in Figure 1. Among them, both the generator and discriminator adopt the idea of progressive generation antagonism [38], and a self-attention mechanism module is introduced into the high-resolution generator and discriminator layer [39]. The network structure of the generator and discriminator is shown in Figure 2. The network starts with a 4 × 4 resolution image and generates 8 × 8 and 16 × 16 resolution images step-by-step until it finally produces a 512 × 512 resolution image. The generator *G* has two functions: one is to generate a pseudo-defect image G(f) based on a true defect-free image *f*, and the other is to generate and reconstruct a pseudo-defect image G(C(i)) based on a pseudo-defect-free image C(i). The generator *C* also has two functions, with one being to generate a pseudo-defect-free image C(i) according to the real defect image *i* and the other being to generate and reconstruct a pseudo-defect-free image C(G(f)) according to the pseudo-defect image G(f). The function of the discriminators $D1_{f}$ and $D2_{f}$ is to distinguish the pseudo-defect-free image C(i) from the real defect-free image *f*, while the function of the discriminators $D1_{i}$ and $D2_{i}$ is to distinguish the pseudo-defect-free image G(f) from the real defect-free image *i*.

The overall optimization goal of the network is to reduce the introduced D2 adversarial loss [40] and cyclic consistency loss [41]. The detailed training process of the DG-GAN is shown in Algorithm 1. The D2 adversarial loss can improve the quality and diversity of the defect images generated by the DG-GAN. The cyclic consistency loss affords the DG-GAN the ability to generate defect images from defect-free images. In order to increase the stability of model training and to improve the quality of generated images, a data augmentation module [14] was introduced at the front-end of the network.
**Algorithm 1** Algorithm training of DG-GAN model.$\theta_{C},\theta_{G},\theta_{D1_{f}},\theta_{D2_{f}},\theta_{D1_{i}},\theta_{D2i}⇐$ Initialize the network parameters**Repeat**Real defect-free image sample $f\in P_{r}\left(f\right)$Real defect image sample $i\in P_{r}\left(i\right)$Obtain generated defect-free image $\hat{f}$ where $\hat{f}=C\left(i\right)$Update discriminator $D1_{f}$’s parameters to maxmize:     $\nabla \theta_{D1_{f}}E_{f\in P_{r}\left(f\right)}\left[logD1_{f}\left(f\right)\right]+E_{i\in P_{r}\left(i\right)}\left[log\left(1-D1_{f}\left(\hat{f}\right)\right)\right]$Update discriminator $D2_{f}$’s parameters to maxmize:     $\nabla \theta_{D2_{f}}E_{i\in P_{r}\left(i\right)}\left[logD2_{f}\left(\hat{f}\right)\right]+E_{f\in P_{r}\left(f\right)}\left[log\left(1-D2_{f}\left(f\right)\right)\right]$Update generator *F*’s parameters to minimize:     $\nabla \theta_{F}E_{i\in P_{r}\left(i\right)}\left[logD1_{f}\left(\hat{f}\right)\right]+E_{i\in P_{r}\left(i\right)}\left[log\left(1-D2_{f}\left(C\left(i\right)\right)\right)\right]+∥G\left(\hat{f}\right)-i∥$Obtain generated defect image $\hat{i}$ where $\hat{i}=G\left(f\right)$Update discriminator $D1_{i}$’s parameters to maxmize:     $\nabla \theta_{D1_{i}}E_{i\in P_{r}\left(i\right)}\left[logD1_{i}\left(i\right)\right]+E_{f\in P_{r}\left(f\right)}\left[log\left(1-D1_{i}\left(\hat{i}\right)\right)\right]$Update discriminator $D2_{i}$’s parameters to maxmize:     $\nabla \theta_{D2_{i}}E_{f\in P_{r}\left(f\right)}\left[logD2_{i}\left(\hat{i}\right)\right]+E_{i\in P_{r}\left(i\right)}\left[log\left(1-D2_{i}\left(i\right)\right)\right]$Update generator *G*’s parameters to minimize:     $\nabla \theta_{G}E_{f\in P_{r}\left(f\right)}\left[logD1_{i}\left(\hat{i}\right)\right]+E_{f\in P_{r}\left(f\right)}\left[log\left(1-D2_{i}\left(G\left(i\right)\right)\right)\right]+∥C\left(\hat{i}\right)-f∥$**Until convergence**Real defect-free image sample $f\in P_{r}\left(f\right)$Obtain generated defect image $\hat{i}$ where $\hat{i}=G\left(f\right)$Return $\hat{i}$

### 3.2. Loss Function

In this paper, D2 adversarial loss [40] and cyclic consistency loss [41] were introduced into the network. The DG-GAN can generate a sufficient number of high-quality and high-diversity defect images based on a small number of defect images.

#### 3.2.1. D2 Adversarial Loss

The DG-GAN network needs to generate high-quality and diverse defective images. However, due to the limited number and diversity of defect images, all feature distributions of defects cannot be covered. To alleviate this problem, D2 adversarial loss was introduced into the network [40]. The difference between D2 adversarial loss and traditional adversarial loss is the addition of a diversity loss. For generator *C* and its corresponding discriminator $D1_{f}$, the adversarial loss is illustrated in Formula (Equation 1):(1)$$\begin{array}{c} L_{gan}\left(C;D1_{f};f;i\right)=E_{\left(f\in P_{r}\left(f\right)\right)}\left[\log D1_{f}\left(i\right)\right]+E_{\left(i\in P_{r}\left(i\right)\right)}\left[\log(1-D1_{f}\left(G(f)\right))\right]. \end{array}$$

In Formula (Equation 1), generator *C* attempts to generate a defect-free image C(i) identical to that in the defect-free domain *F* so that discriminator $D1_{f}$ considers the generated defect-free image to be true, even if the value of $D1_{f}\left(C\left(i\right)\right)$ is close to 1, which minimizes $L_{gan}\left(C;D1_{f};f;i\right)$.

In the GAN network training process, optimize the $P_{g}$ distribution of the generated image to reduce the JS divergence $D_{JS}\left(P_{r}\left|\right|P_{g}\right)$ between the real image distribution $P_{r}$ and the generated image distribution $P_{g}$. It has been proven that using the KL divergence $D_{KL}\left(P_{g}\left|\right|P_{r}\right)$ generation model generated sample image is of high quality but lacks diversity [42]. To mitigate the above problems, two discriminators, $D2_{i}$ and $D2_{p}$, were added to the base network to increase the diversity of image defects generated. In contrast to discriminators $D1_{i}$ and $D1_{p}$, $D2_{i}$ and $D2_{p}$ have an output of 1 for the generated image and 0 for the real image. The loss function is illustrated in the Formulas (Equation 2) and (Equation 3):(2)$$\begin{array}{c} L_{gan2}\left(G;D2_{i};f;i\right)=E_{\left(f\in P_{r}\left(f\right)\right)}\left[\log D2_{i}\left(G\left(f\right)\right)\right]+E_{\left(i\in P_{r}\left(i\right)\right)}\left[\log\left(1-D2_{i}\left(i\right)\right)\right] \end{array}$$
(3)$$\begin{array}{c} L_{D2gan}\left(G;D1_{i},D2_{i};f,i\right)=L_{gan}\left(G;D1_{i};f;i\right)+\lambda_{1}L_{gan2}\left(G;D2_{i};f;i\right). \end{array}$$

In Formulas (Equation 2) and (Equation 3), $\lambda_{1}$ controls the balance between similarity and diversity, $L_{gan2}\left(G;D2_{i};f;i\right)$ ensures the diversity of defect images generated; $L_{gan}\left(G;D1_{i};f;i\right)$ computes the quality of defect images generated; and $L_{D2gan}\left(G;D1_{i},D2_{i};f,i\right)$ balances the quality and variety of defect images generated.

#### 3.2.2. Cycle Consistency Loss

In actual production, it is easier to obtain a large number of defect-free images, because real defect images are difficult to collect. In addition, there is no difference between a defect image and a defect-free image, except for the defective area. Therefore, the best method for generating defect images is based on defect-free images to assist in defect generation, rather than directly generating defect images [42]. In order to realize the function of defect-free-image-assisted defect image generation, this paper introduced cyclic consistency loss into the network, as shown in Formula (Equation 4):(4)$$\begin{array}{c} L_{cyc}\left(G,C\right)=E_{\left(f\in P_{r}\left(f\right)\right)}\left\|C\left(G\left(f\right)\right)-f{\|}_{1}+E_{\left(i\in P_{r}\left(i\right)\right)}\right\|G\left(C\left(i\right)\right)-i{\|}_{1}. \end{array}$$

In Formula (Equation 4), for the generator *C* and the defect image G(f) generated by *G* are taken as inputs to generate the reconstructed pseudo-defect-free image C(G(f)), which is close to the real defect-free image *f*, where the measure is the L1 norm. Similarly, for generator *G*, the defect-free image C(i) generated by *C* is taken as input to reconstruct and generate the pseudo-defect image G(C(i)), which is similar to the real defect image *i*. The reconstructed defect-free image C(G(F)) is finally similar to the input defect-free image *f*, and, as a result, the generated defect-free image G(f) maintains its similarity to the input defect-free image *f* in the defect-free region. By using cyclic consistency loss, DG-GAN can preserve the common features of both defect images and defect-free images.

### 3.3. Self-Attention Mechanism Module

Traditional Generative Adversarial Networks become unstable when generating images above 256 × 256 resolution, but perform optimally when generating low-resolution images. There are obvious differences between the generated data and the actual data, and the network loss curve obviously fluctuates. This is due to the network’s limited grasp of remote context information and the lack of image details, which magnifies the difference between the generated data and the real data. Although the traditional convolutional algorithm has irreplaceable advantages in processing local domain information, it has obvious shortcomings in processing remote correlations [43]. Introducing self-attention mechanisms [39] into the network can enhance the network’s ability to extract remote context information, coordinate the correlation details between each pixel position, and increase the importance of basic feature information. Therefore, the self-attention mechanism was introduced into the DG-GAN network and applied to the 256 × 256 and 512 × 512 resolution generators and discriminator layers to enhance the network’s feature extraction capabilities and training stability. The network structure of the self-attention mechanism is illustrated in Figure 3.

The definition of the self-attention mechanism is illustrated in the Formulas (Equation 5)–(Equation 10). Assuming that the input data to the attention module from the previous feature layer are *X*, then the attention degree $\xi_{j,i}$ of certain position *i* to position *j* is shown in Formula (Equation 5).
(5)$$\xi_{j,i}=SoftMax\left(Q_{i}^{T}K_{j}\right)=\frac{exp\left(Q_{i}^{T}K_{j}\right)}{\sum_{i=1}^{N}exp\left(Q_{i}^{T}K_{j}\right)}.$$

The output of the self-attention layer is shown in Formula (Equation 6):(6)$$S_{j}=W_{S}\left(\sum_{i=1}^{N}\xi_{j,i}V_{i}\right).$$

Among them: (7)$$\begin{array}{c} Q=W_{Q}X\in R^{H\times W\times C1} \end{array}$$(8)$$\begin{array}{c} K=W_{K}X\in R^{H\times W\times C2} \end{array}$$(9)$$\begin{array}{c} V=W_{V}X\in R^{H\times W\times C3}. \end{array}$$

In Formulas (Equation 6)–(Equation 9), $W_{S}$, $W_{Q}$, $W_{K}$, and $W_{V}$ represent the learnable convolution kernels of 1 × 1, whose functions are to change the number of channels of input *X*; *S*, *Q*, *K*, and *V* represent the respective outputs after the operation; *H* and *W* represent the height and width of the input and output, respectively; and C1, C2, and C3 are the number of channels.

Finally, combined with the output *S* of the attention layer and the input *X* of the feature layer, an auxiliary hyper-parameter η growing from 0 to 1 was set so that the learning of the model could gradually expand from local features to the whole. The final output *Y* is presented in Formula (Equation 10):(10)Y=X+ηS.

### 3.4. Data Augmentation Module

When a limited amount of data is used for image generation, overfitting easily occurs in network training [44], which makes it difficult to converge the generated model and ultimately leads to obtaining low-quality generated images, which affect the development of subsequent detection tasks. To solve this problem, this paper introduced a data augmentation module [14] at the front end of the DG-GAN network to augment the number of original real defect images and ensure that they did not affect the data distribution of the generated images. Three fixed data enhancement operations were used in this paper: image scaling, image rotation, and mirroring.

## 4. Experiments

This section first introduces the dataset and experimental parameter settings used in the experiment. Secondly, the evaluation method used in the experiment is briefly introduced. Then, all the methods used in this paper were performed by ablation experiments, and the performance scores before and after were compared to determine the effectiveness of the methods and their contribution. Finally, in order to determine the time cost of the model training, the training times of ProGAN [38], CycleGAN [45] and DG-GAN were compared.

### 4.1. Implementation Details

To verify the validity of the defect image generation method proposed in this paper, the NEU public dataset [46] and IP-def dataset were used for ablation experiments. The NEU dataset contains a total of six typical types of surface defects in a hot-rolled steel strip, namely, Rolled-in Scale (RS), Patches (Pa), Crazing (Cr), Pitted Surface (PS), Inclusion (In), and Scratches (Sc). There are 300 defect images for each type, and the image size is 200 × 200. The dataset was manually marked with the defect location and category. All experiments conducted in this paper were based on PyTorch [47] frameworks within the Windows environment using three Titan-X 12GB graphics cards and CUDA version 11.5. The initial input to the generator was a 512-dimensional random vector that followed a normal distribution. We set the batch size of the model to eight. In this experiment, an Adam Optimizer [48] was used, and the training parameters were set as $\beta_{1}$ = 0.9, $\beta_{2}$ = 0.999, and ε = $10^{-8}$. Finally, the learning rate for the generator and discriminator was set to 0.001, and the number of epochs was set to 1000.

### 4.2. Evaluation Index

As we all know, the evaluation of generation models is a major problem facing today’s scientific researchers [49]. At present, mainstream image quality assessment methods mainly include the Inception Score (IS) [50] and the Fréchet Inception Distance(FID) [51]. The FID measures the similarity between two sets of images from the statistical aspects of the computer vision features of the original image and is a measure of the distance between the feature vectors that calculate the real image and the generated image. The lower the FID score, the more similar the two sets of images are, or the more similar their statistics are. Among them, the IS has certain limitations. It is difficult to assess the authenticity and diversity of the image details generated. The FID is more authoritative than the IS in assessing the quality of generated images, and its evaluation results are more similar to human evaluation results. Therefore, the FID was used to evaluate the diversity and similarity of the images generated. In this paper, the FID score was introduced to evaluate the quality of the generated image in the experiment. The lower the FID score, the better the quality of the generated image, and the closer the data distribution was to the real image.

### 4.3. Ablation Study

In this paper, the generation performance of the improved network was compared with that of the Pro-GAN [38] and CycleGAN [45]. Figure 4 and Figure 5 show the visualized results of the generated experiments on the NEU and IP-def datasets, respectively. Table 3, through Rows 1, 2, and 10, presents a comparison of the FID scores for the three generation methods. In this paper, three strategies were adopted to improve the network model. Firstly, the D2 adversarial loss function and the cyclic consistency loss function were introduced; then, the data augmentation module was entered into the front-end of the network, and, finally, we introduced the self-attention mechanism module into the high-resolution layer of the generator and discriminator. As shown in Table 3, the FID score for the generated defect image for each category, when each module was absent, was calculated independently.

#### 4.3.1. Loss Function

This section explores the influence of loss function substitution on the DG-GAN network performance before and after implementation. By introducing the D2 adversarial loss function and the cyclic consistency loss function, the FID score of the generated images was compared with those before, and conclusions could be drawn. As shown in the second and fourth lines of Table 3, after the introduction of the above two loss functions into the network, the FID score of the generated image significantly decreased, thus indicating that the quality of the generated image was significantly improved. Since D2 adversarial loss is a diversity loss added on the basis of the original adversarial loss, its introduction into the network can improve the diversity of the generated images, while cyclic consistency loss can realize the generation of defect images from defect-free image samples, and the generated images have high quality and diversity. Therefore, D2 adversarial loss and cyclic consistency loss were introduced in this paper to improve network generation performance.

#### 4.3.2. Data Augmentation

Due to the limited number of defect image samples in the original dataset, direct input to the network for training was not effective. Therefore, the data augmentation module was introduced at the DG-GAN’s front end to extend the dataset to at least three times the original. Experiments show that the augmented dataset achieved better training results, as shown in the rows 4 and 10 of Table 3. As shown in Figure 6, the loss curve of the network after data augmentation was more stable than that of the previous network.

#### 4.3.3. Self-Attention Mechanism

This section explores the impact of the self-attention mechanism on the performance of the DG-GANs. The self-attention mechanism modules were incorporated into the middle layer of the generator and discriminator, respectively. By comparing the FID scores of the generated images, conclusions were drawn. As shown in lines 4 to 9 of Table 3, the introduction of the self-attention modules into the low-resolution layer had little or no impact on quality of the generated image. However, the introduction of the self-attention modules into the 256 × 256 resolution layer and above led to a significant improvement in the quality of the generated images. Because convolution operators have local acceptance domains, it is difficult to find the remote dependencies of small convolution kernels on high-resolution images, while self-attention is more effective at establishing remote dependencies [43]. Therefore, the self-attention mechanism was adopted in the 256 × 256 and 512 × 512 resolution layers of the generator and discriminator to improve the performance of network generation.

### 4.4. Time Comparison

The training time cost of the defect generation model is an important index to consider. The training time of the defect generation model is closely related to the network structure, the number of parameters, the number of training iterations, and the size of the data set. This paper compared the training times of ProGAN, CycleGAN, and DG-GAN, as shown in Table 4. In the table, the training time is the sum of the time spent on the generation experiments of the two datasets. Although the DG-GAN’s parameters were larger than the other models and the training time was longer than the other two models, the DG-GAN’s training time was completely acceptable. In addition, the DG-GAN could generate defective images of higher quality and diversity than the previous two methods.

## 5. Case Study of Defect Detection

In order to verify the validity of the generated pseudo-defect image in the task of defect detection, this paper used the DG-GAN to generate an augmented dataset of pseudo-defect images under two different datasets for defect detection experiments. The experiment was mainly divided into two parts, including the part where the DG-GAN was generated into defect images and the part where the YOLOX detector [52] was used for defect detection. The overall process is shown in Figure 7. The YOLOX is a new high performance detector. It switches the YOLO detector to anchor-free mode and introduces the decoupling head, as well as SimOTA’s leading label allocation strategy, to achieve the most advanced results in the experiment. It can ensure high detection accuracy while achieving good detection speed to meet the requirements of real-time detection. In this paper, the total number of iterations of the detector was set to 400, the batch size was set to 16, the SGD optimizer was used, the weight decay was 0.0005, the SGD momentum was 0.9, the initial learning rate was 0.01, the cosine annealing algorithm was used to dynamically update the learning rate, and the final learning rate was 0.001.

### 5.1. Case 1: Surface Defect Detection of Hot-Rolled Strip Steel

Hot-rolled strip steel is widely used in the automobile, electric machinery, chemical industry, shipbuilding, and other industrial sectors, but also in cold-rolled, welded pipe, and cold-formed steel blank production. However, due to environmental factors, raw material composition, production technology, and uncontrollable human factors, there are often various defects in the production and manufacture of hot-rolled strip steel, including transportation, storage, and actual use. These defects may be large or small, but they will affect the performance and service life of the hot-rolled strip, thereby ultimately affecting the safety of use. The stability and integrity of the hot-pressed steel strip is of great significance to the safety of production.

The dataset used in the experiment was the surface defect dataset of hot-rolled strips produced by Northeastern University, called NEU, which includes six types of surface defects on hot-rolled strips. The characteristic of defect image samples is that the differences within the class are relatively large, and the size, position, and shape of the same type of defect are not fixed. In addition, the differences between some defects and the background are not obvious. A total of 5400 false defect images were generated using the DG-GAN, and all false defect images were manually labeled.

This paper employed the true original defect image and a varying number of generated false defect images to train and compare a detection model. The final training loss and validation loss of different training sets are shown in Figure 8. It can be observed that using more pseudo-defect images could make the training more stable, and the use of more datasets generating pseudo-defect images could alleviate the overfitting phenomenon of the model and make the training of the model more stable. The first row of Table 5 shows the detection accuracy index under different numbers of NEU-generated datasets, thus demonstrating that using more generated data can improve the detection accuracy of the model.

### 5.2. Case 2: Detecting Surface Defects of an Insulation Partition

The insulation partition is made of insulating material, which is used to isolate electrical parts, restrict the movement of personnel, and prevent access to high-voltage live parts of the insulation plate. The insulation partition, also known as the insulation baffle, should generally have high insulation performance. It can be connected directly to the live part of 35kV and below, thereby acting as a temporary shield. As one of the most commonly used components in electric power operation, it plays an irreplaceable role in the safety specification of electric power production and transportation. However, according to the investigation, various defects, such as scratches, pit points, ablation, and voltage breakdown will inevitably occur during the process of manufacturing, transportation, storage, and use [20]. These defects pose a potential threat to the normal development of electric power production and the life safety of electric power operators, and they may lead to major safety accidents. Therefore, it is very important to detect the insulation partition for defects.

In this experiment, the IP-def insulation partition defect dataset was collected through investigations of both production and sales manufacturers and power companies. There are four most common types of defects identified in the IP-def, including Scratches (IP-SC), Pit Points (PPs), Ablation (Ab), and Voltage Breakdown (VB). Each type of defect is independent of the other, and is captured by the phone’s camera and saved as a jpg image. The size of each defective image is 512 × 512, and there are 100 defective images in each type, with 400 in total. Figure 9 shows an image of a normal insulating partition and an image of the four types of defects. In the experiment, the DG-GAN was first used to expand the dataset, and 2000 defect images were obtained to expand the dataset. Then, the original dataset, Fake1000 dataset, and Fake2000 dataset were correspondingly used to train and verify the detection model. The final training and verification losses are shown in Figure 10, and the detection accuracy is shown in line 2 of Table 5. This experiment demonstrates that using more generated pseudo-defect images to train the model can improve the performance of the defect detection model.

## 6. Conclusions

In this paper, a defect image generation network DG-GAN was proposed to solve the problem of insufficient defect image samples in defect detection tasks. Based on the Pro-GAN, data augmentation and self-attention mechanism modules were introduced in this network, and the D2 adversarial loss function and cyclic consistency loss function were adopted. The network could generate high-quality and high-diversity defect images, expand the original defect data set, ease the difficulty of defect image sample collection, and lay a solid foundation for the success of defect detection tasks. In this paper, the NEU dataset and IP-def dataset were also used to verify the validity of the DG-GAN. By augmenting the dataset, the detection accuracy of the YOLOX detector was improved.

In the defect generation experiment, the DG-GAN achieved lower FID scores than state-of-the-art methods, with a mean reduction of 16.17 and 20.06, respectively, and a maximum reduction of 32.5 and 28.1, respectively, thereby resulting in higher-quality and a more diverse range of defect images. In two instances of defect detection, the accuracy of detection before and after augmenting the two datasets was compared. In the NEU dataset, the YOLOX detection accuracy increased from 72.1% to 78.2%, and in the IP-def dataset, the YOLOX detection accuracy increased from 75.2% to 95.6%. The above results demonstrate that the DG-GAN model achieves excellent performance in defect image generation, thus laying a strong foundation for future defect detection tasks. Despite the remarkable progress made by the DG-GAN, it still possesses certain shortcomings. For instance, the training time of the DG-GAN is relatively lengthy. To achieve cost control objectives, the complexity of the network can be reduced, the number of network parameters can be reduced, and lightweight networks can be established. The training time can be reduced without affecting the model generation performance.

## Figures and Tables

**Figure 1 sensors-23-05922-f001:**
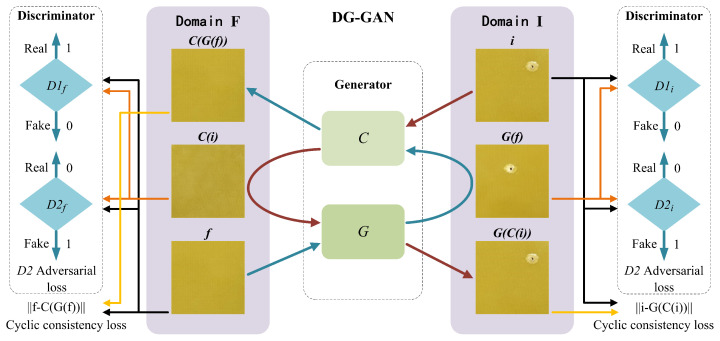
Network structure of DG-GAN: it consists of two generators (*C*, *G*) and four discriminators ($D1_{f}$, $D1_{i}$, $D2_{f}$, $D2_{i}$). The goal is to optimize D2 adversarial loss and cyclic consistency loss. *f*, C(i), and C(G(f)) represent defect-free images at different stages, and *i*, G(f), and G(C(i)) represent defect images at different stages.

**Figure 2 sensors-23-05922-f002:**
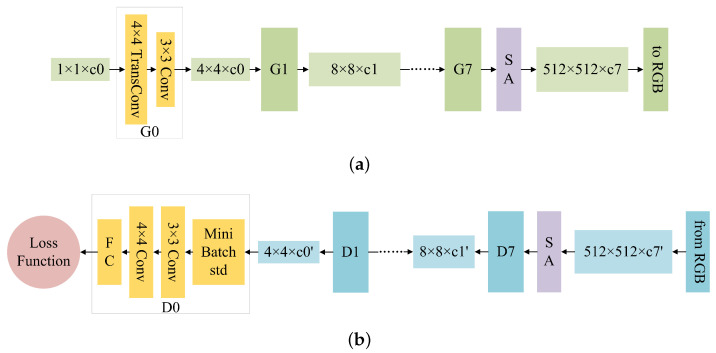
Network structure of generator and discriminator of DG-GAN. (**a**) Structure of generator. (**b**) Structure of the discriminator.

**Figure 3 sensors-23-05922-f003:**
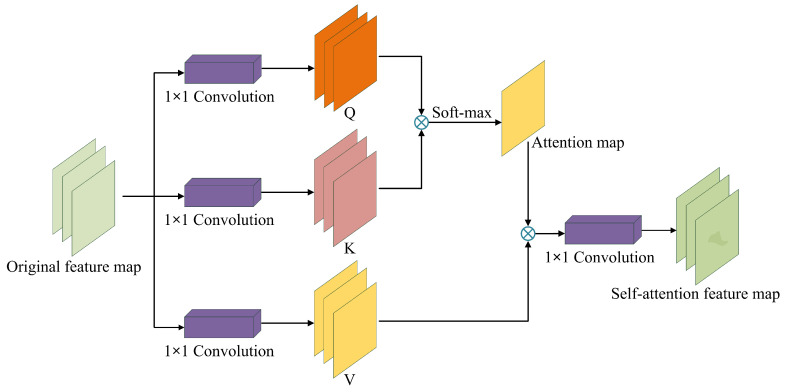
Network structure of self-attention mechanism modules; ⊗ stands for matrix multiplication, *Q*, *K*, and *V* are three weight matrices, which respectively represent the query matrix, the key matrix, and the value matrix.

**Figure 4 sensors-23-05922-f004:**
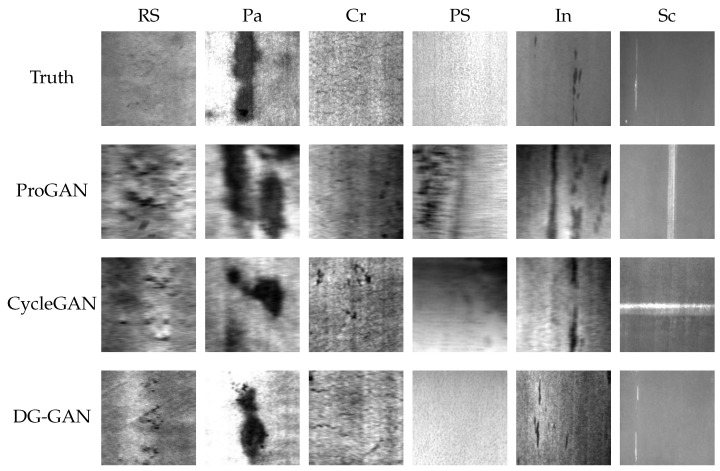
Visualization of the generated images of the three generation models in the NEU dataset.

**Figure 5 sensors-23-05922-f005:**
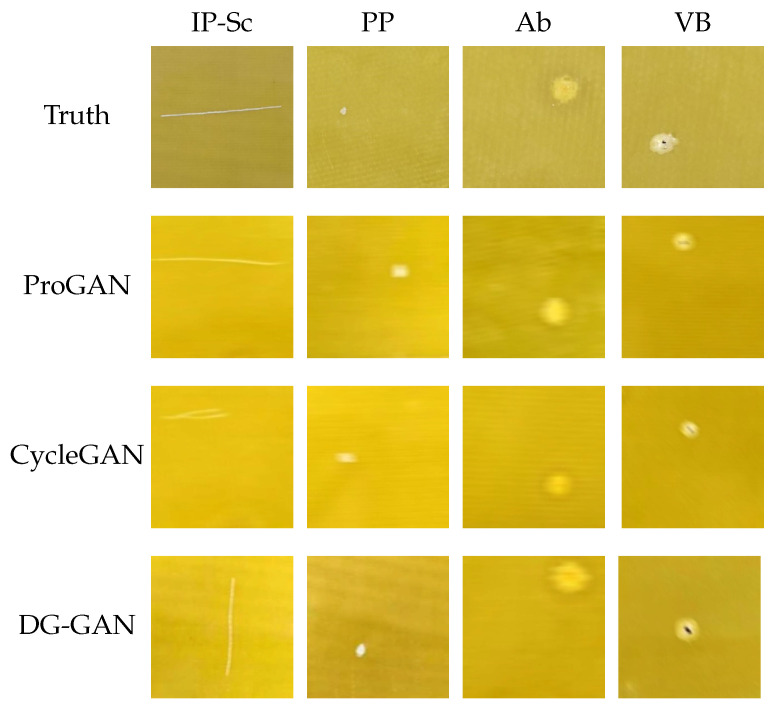
Visualization of the generated images of the three generation models in the IP-def dataset.

**Figure 6 sensors-23-05922-f006:**
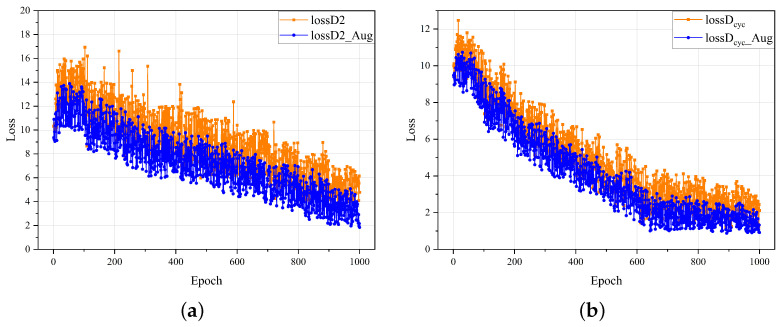
Comparison of training losses in DG-GAN without using data augmentation modules versus using data augmentation modules. (**a**) D2 Adversarial loss curve. (**b**) Cyclic consistency loss curve.

**Figure 7 sensors-23-05922-f007:**
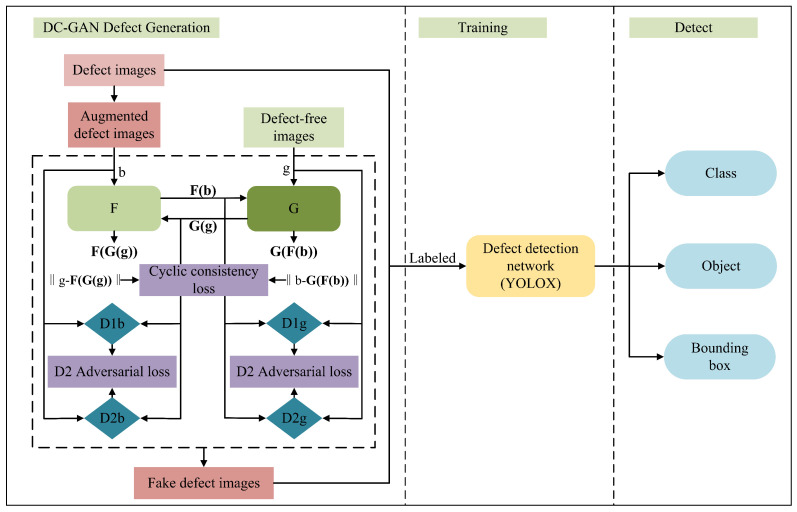
Overall process of defect image generation and detection.

**Figure 8 sensors-23-05922-f008:**
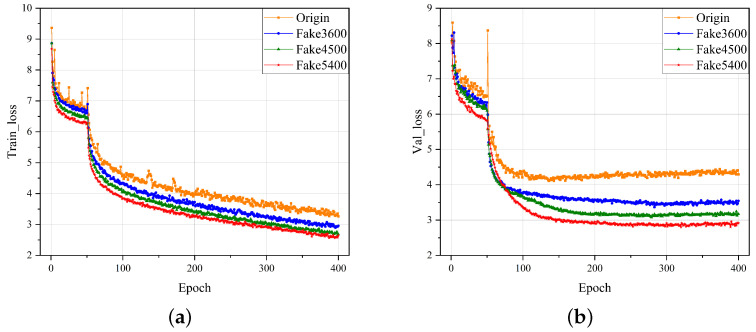
Loss of training and validation of NEU datasets using real defect images and varying numbers of generated defect images. (**a**) Training loss curve. (**b**) Validation loss curve.

**Figure 9 sensors-23-05922-f009:**
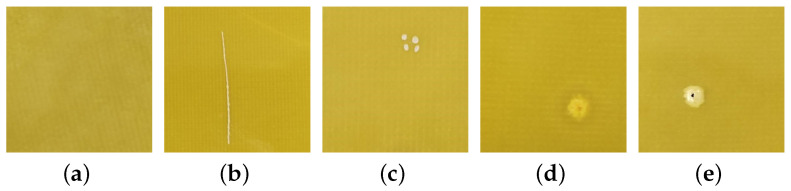
Normal insulation partition image and insulation partition defect image. (**a**) Normal. (**b**) IP-SC. (**c**) PP. (**d**) Ab. (**e**) VB.

**Figure 10 sensors-23-05922-f010:**
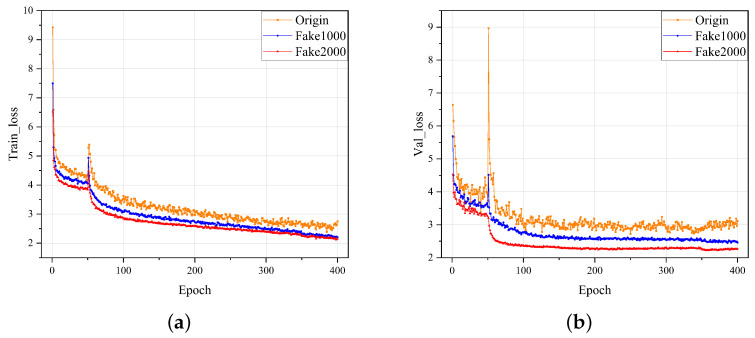
Loss of training and validation of IP-def datasets using real defect images and varying numbers of generated defect images. (**a**) Training loss curve. (**b**) Validation loss curve.

**Table 1 sensors-23-05922-t001:** Comparison of advantages and disadvantages of some typical generation methods.

Methods	Improved Point	Advantage	Insufficient
CGAN	Adds constraints	Increased control	Training instability
DCGAN	Convolutional structure introduced	Improved universality and stability	The optimization method does not improve
WGAN	Introduces Wasserstein distance	Improved training stability and generation diversity	The poor quality of generated samples
CycleGAN	Introduces a loop mechanism	Low sample requirement	The generated image quality is poor
ProGAN	Asymptotically generates adversarial ideas	Training stability	Long training time

**Table 2 sensors-23-05922-t002:** Comparison of some classical defect detection models. mAP represents the average detection accuracy of the model for the PASCAL VOC 2007 dataset and the PASCAL VOC 2012 dataset.

Model	Improvement	Insufficient	mAP
R-CNN	mAP greatly improved; introduced RP+CNN.	The training steps are tedious, and the training speed is slow.	58.50% 62.00%
Fast R-CNN	mAP has been improved, and training time has been shortened.	Unable to meet real-time requirements, and unable to detect end-to-end.	70.00% 66.00%
Faster R-CNN	Both accuracy and speed are improved, thus enabling end-to-end detection.	Real-time detection cannot be achieved, and the calculation is large.	73.20% 70.40%
YOLOv4	Large residual blocks, SPP, and PANNnet are introduced, and Mish activation functions are used.	Small target detection is poor, and the recall rate is relatively low.	85.46% 86.68%
YOLOv5	Adaptive anchor calculation, focus structure, and CSP structure are introduced, and GIOU_Loss is adopted.	Small target recognition is unstable and requires massive training data.	89.10% 92.68%
SSD	Multiscale detection, default anchor, data enhancement.	Needs artificial setting box value, and small target recognition effect is poor.	76.90% 77.91%

**Table 3 sensors-23-05922-t003:** Among the image quality evaluation indexes generated under different model structures, bold text indicates the index with a better score under current comparison. Base represents a network with an improved loss function, and Base+SA(X) represents the addition of a self-attention mechanism module to the X-resolution layer of the base network.

Model∖FID∖Class	RS	Pa	Cr	PS	In	Sc	IP-Sc	PP	Ab	VB
Pro-GAN	64.9	57.8	46.5	51.2	43.6	46.2	43.1	42.5	40.3	48.5
CycleGAN	60.5	63.2	50.1	49.3	56.5	49.6	45.5	46.7	48.6	53.5
Base	**40.2**	**43.1**	**37.6**	**43.8**	**30.8**	**39.4**	**35.2**	**34.3**	**35.1**	**40.4**
Base+SA(8)	45.9	45.3	38.2	43.5	32.9	42.2	34.8	36.9	37.5	42.7
Base+SA(32)	44.6	45.8	39.4	43.9	33.1	43.4	35.6	35.8	36.9	43.1
Base+SA(128)	43.2	44.3	37.9	45.2	31.9	41.6	36.7	34.8	36.2	41.8
Base+SA(256)	**38.8**	**41.9**	38.5	**40.3**	32.5	**38.8**	**32.6**	35.3	**33.2**	42.5
Base+SA(512)	**38.3**	42.3	**36.6**	41.6	**30.1**	**38.1**	33.7	**32.4**	33.9	**38.6**
Base+Aug	**36.9**	**40.1**	37.8	42.1	**29.2**	**37.2**	**32.3**	**30.9**	34.5	**35.9**
DG-GAN	**32.4**	**36.6**	**34.3**	**38.2**	**28.4**	**30.6**	**29.9**	**28.1**	**29.6**	**31.8**

**Table 4 sensors-23-05922-t004:** Comparison of training times of various image generation methods.

Method	Variable Size (Mb)	Epoch	Data Volume	Training Time (h)
ProGAN	46.2	1000	2200	8.15
CycleGAN	35.30	1000	2200	6.53
DG-GAN	48.3	1000	2200	8.32

**Table 5 sensors-23-05922-t005:** Detection accuracy of NEU datasets and IP-def datasets using different datasets.

Dataset	mAP	Origin	Fake1000	Fake3600	Fake2000	Fake4500	Fake5400
NEU	0.5	72.1	-	74.3	-	77.3	78.2
	0.5:0.95	38.6	-	40.7	-	43.5	45.4
IP-Def	0.5	75.2	86.3	-	95.6	-	-
	0.5:0.95	39.4	63.2	-	75.8	-	-

## Data Availability

I am sorry that the data used by the research institute cannot be disclosed due to the confidentiality requested by the project team.

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
