# Peer review of "DG-GAN: A High Quality Defect Image Generation Method for Defect Detection"

_sensors, 2023, doi:10.3390/s23135922_

Round 1

Reviewer 1 Report

The manuscript proposes a novel defect image generation method named DG-GAN. The proposed DG-GAN is different from previous generation methods, and DG-GAN can achieve promising solution for defect detection. My main concerns are given below.

1. The main experimental results should be contained in Abstract.

2. In Introduction, the authors introdcued previous generation models and analyzed their problems. Then the authors presented DG-GAN,  i.e., "the  idea of progressive generation antagonism is used to improve the training stability of the network, and D2 adversarial loss function, cyclic consistency loss function, data enhancement module and self-attention mechanism are used to improve the generation effect of the network, and finally to generate defect images with high quality and high diversity. " However, the motivation of the proposed is not clear. In other words, why did the authors use these techniques to build the model that would solve the problems of previous approaches?

3. Related work. The authors are suggested to analyze the problems of previous work at the end of related work, and then give the difference between DG-GAN and previous method, enhancing the novelty and contribution of the manuscript.

4. Method part:

1) The resolution of Figures is very low, which decreases the quality of the manuscript.

2) Adding a algorithm describing DG-GAN can increase the integrity and readability of the article.

5. Experiments part: 

1) GG-GAN should be compared with state-of-the-art generation methods to demonstrate its superority.

2) The time cost is an important metric for the proposed method. However, it is missing.

6. The summarize of this paper is too weak, should be strenthed. The results obtained must be clearly stated. The effectiveness of the method is confirmed by digital material. In addition, the shortcomings of the article should also be summarized.

Reviewer 2 Report

This paper presents a reasonable method to solve a real application problem. It is well-organized, clearly writing, and shows some interesting results that encouraged to be accepted with major revision. However, the commented questions need only to be answered.

1.    Please explicitly indicate and clarify the challenges this study aims to address. What are the challenges and why? Why cannot the previous studies well address these challenges. 

2.  At the end of section 1 add a table that summarizes the advantages and disadvantages of existing methods facing the same problem. This way the reader would rapidly appreciate novelty of the paper.

3. Please enrich the captions of all figures and tables for clarification.

4.    Why do you select self-attention? There are many state of the arts attention strategies that used with more popular and demonstrate better performance.

5.  In the comparison to SOTA methods, more experimental results of other state-of-the-art methods should be given.

6.    I also find some grammar problems in this paper. Author needs to carefully check these low mistakes, which is very important for readers.

This paper presents a reasonable method to solve a real application problem. It is well-organized, clearly writing, and shows some interesting results that encouraged to be accepted with major revision. However, the commented questions need only to be answered.

1.    Please explicitly indicate and clarify the challenges this study aims to address. What are the challenges and why? Why cannot the previous studies well address these challenges. 

2.  At the end of section 1 add a table that summarizes the advantages and disadvantages of existing methods facing the same problem. This way the reader would rapidly appreciate novelty of the paper.

3. Please enrich the captions of all figures and tables for clarification.

4.    Why do you select self-attention? There are many state of the arts attention strategies that used with more popular and demonstrate better performance.

5.  In the comparison to SOTA methods, more experimental results of other state-of-the-art methods should be given.

6.    I also find some grammar problems in this paper. Author needs to carefully check these low mistakes, which is very important for readers.

Reviewer 3 Report

Title: DG-GAN: A high quality defect image generation method for defect detection

The manuscript introduces a defect image generation method for defect detection. However, the following should be rectified before acceptance.

·        Plagiarism excluding the bibliography is 37%. It needs to be minimized with the journal threshold.

·        Illustrations using figures are needed to emphasize the work. Only results are illustrated.

·        There are typos and grammatical errors which should be rectified.

·        It would be better to include the existing defect detection algorithms and their performance in a tabular form for a clear visibility.

·        Table 1 needs a revision. What does the first column title state?

·        Section 4.3.2-Data Enhancement or Data Augmentation?

Enhancement usually refers the process of quality improvement using some filters.

·        The term FID is not mentioned earlier in the text. What does FID state?

·        It is better to illustrate different types of defects on the objects using figures. Visualize them before and after detection and generation.

·        Section 5 can be Conclusion instead of Summarize.

Need a proof reading and re-writing of some sentences.

Reviewer 4 Report

This paper has complete structure and clear description of the motivation of this research. The main methodology is also described  relatively clear.

However, there are some details can be improved:

1. Add more description in figure captions. For example, Figrue1 and Figure 3's captions can include the notes of certain letters/phrase, such as C, G, or Q,K, V. 

2. In section 4.2, FID is used first time, but there is no full name of it.

3. Table 1, may need explain a little of each model used in comparison, like what is Base +SA(XX)

4. May create a separate section for case study.

5. May apply proposed method on detection with multiple network, not only one YOLOX can provide more robust proof.

6. The GAN generated data can help increase the performance of detection should be used in many cases. Then what is this unique contribution of this work? May considering use other GAN models (mentioned in literature review ) as a base line, too.

6. In literature review, why there is no accuracy/performance of cited GAN relative defect detection works.

Use too many long complex sentences which makes reader lose the track.

Round 2

Reviewer 2 Report

The manuscript is well-revised, and it is acceptable in its current form.

The manuscript is well-revised, and it is acceptable in its current form.

Author Response

Thank you very much for your comments and suggestions, which are of great help to improve the quality of this paper.

Reviewer 4 Report

Thanks for your responses. The newer version looks much better. Here a few following comments of current version:

1. For Table 3, may add a tab space before each equation at each step, which will make the algorithm more clear. i.e. "Update discriminator D’s parameters to maxmize:\n <tab> equation"

2. The paper means to propose a new GAN structure to improve the small target defect detection. The GAN relative methodology and results are very clear and make sense. However, it is not sufficient that there is only one defect detection method involved to illustrate the proposed GAN's improvement in defect detection. If there is difficulty to show another detection result, how about the similar and simpler YOLO or YOLO v4/v5  + your DG-GAN to show the more general proof of the improvement.

3. please clean the appendix parts in the end, like the data availability etc, if the items are not applied for you, please remove it.  Especially the abbreviations section, may consider update it to fit your manuscript.

N/A
